# OPENESTIMATE: EVALUATING LLMS ON REASONING UNDER UNCERTAINTY WITH REAL-WORLD DATA

**Alana Renda, Jacob Andreas, Jillian Ross**
MIT CSAIL
{marzoev,jda,jillianr}@mit.edu

## ABSTRACT

Real-world settings where language models (LMs) are deployed — in domains spanning healthcare, finance, and other forms of knowledge work — require models to grapple with incomplete information and reason under uncertainty. Yet most LM evaluations focus on problems with well-defined answers and success criteria. This gap exists in part because natural problems involving uncertainty are difficult to construct: given that LMs have access to most of the same knowledge as humans, it is non-trivial to design questions for which LMs will struggle to produce correct answers. As a result, LM performance on reasoning under uncertainty remains poorly characterized. To address this gap, we introduce OPENESTIMATE, an extensible, multi-domain benchmark for evaluating LMs on probabilistic estimation tasks that require models to synthesize knowledge from pretraining and express predictions as Bayesian priors. We assess these priors for accuracy and calibration. Across six frontier models, we find that LM-elicited priors are worth the equivalent of about five samples from the underlying data distribution, and that posteriors computed using LM priors tend to be more accurate than those computed using a naive prior. At the same time, the relationship between model accuracy and confidence is weak across the board, indicating the value of developing new methods to improve calibration. The OPENESTIMATE benchmark thus offers a challenging evaluation for frontier LMs and a platform for developing models that are better at probabilistic estimation and reasoning under uncertainty.

## 1 INTRODUCTION

Language models (LMs) have demonstrated strong performance across a broad range of reasoning tasks. However, most existing evaluations are confined to problems with clearly defined answers and assume access to complete, unambiguous information. In contrast, many real-world applications in which LMs are deployed are characterized by open-endedness and uncertainty.

For example, consider a financial analyst assessing the total addressable market of an early-stage investment. To perform this task, the analyst must draw on accumulated knowledge of the competitive landscape and industry dynamics to construct a coherent model of the domain and produce an informed initial estimate. Because the setting is inherently uncertain, the analyst's beliefs are best expressed not as a point estimate but as a probability distribution over possible outcomes—in Bayesian terms, a *prior*—which captures both a central estimate and the analyst's confidence in it. Generating sound priors requires not only probabilistic reasoning skills, but also the ability to synthesize heterogeneous, noisy, and sometimes opaque sources of evidence into a structured format for downstream inference.

This use case is not unique in its requirements—analogous problems exists across a variety of domains, including healthcare, public policy, and scientific discovery. However, despite the ubiquity of applications involving uncertainty, existing benchmarks seldom test models on their ability to generate accurate and well-calibrated Bayesian priors in realistic contexts. Some past work (Xia et al., 2024; Wong et al., 2025) has studied procedures for eliciting probabilistic models from LMs, but most specify the task as a mathematical exercise with fully specified inputs (Paruchuri et al., 2024), or as forecasting questions that are time-bounded and whose outcomes eventually leak into training data (Karger et al., 2024).

Designing a faithful evaluation of this capability is nontrivial. A good benchmark must be grounded: it should require the model to draw on background knowledge from pretraining to form high-quality

priors. At the same time, it must avoid information leakage: if the ground-truth answer already exists in the training data, the benchmark tests memorization rather than reasoning.

To address this gap, we introduce an evaluation procedure based on *derived conditional statistics*: summary statistics computed by filtering large, public observational datasets using randomly sampled conditions. Concretely, we ask the model to estimate the true value of a statistic derived from a public dataset—for example, "the average funding raised by non-tech companies outside the US with more than 10 employees" (from Pitchbook (PitchBook Data, 2024)), or "the average weight of US adults with diabetes and blood mercury levels within a specified range" (from NHANES (Centers for Disease Control and Prevention, 2018)). Because the filtering criteria (conditions) used to derive these statistics are drawn randomly from a large space of possible attribute combinations, the resulting quantities are empirically verifiable yet unlikely to appear verbatim in pretraining corpora.

We use this procedure to construct OPENESTIMATE, a benchmark for evaluating LMs' probabilistic estimation capabilities. The benchmark comprises of 178 such statistics across three domains and can be readily extended to new ones without labor-intensive data collection. Models are presented with natural language descriptions of these statistics and are asked to express their estimates of the true value of the statistics as Bayesian priors, which are evaluated in terms of accuracy (whether predicted distributions concentrate near the ground truth) and calibration (whether stated confidence levels align with observed frequencies).

Using OPENESTIMATE, we evaluate the quality of priors elicited from frontier LMs, and find that these models are far from omniscient: in terms of accuracy and calibration, they often perform no better—and often worse—than estimates derived from only a handful of samples from the underlying population. At the same time, these priors could still be useful in practice: posteriors computed using LM priors tend to be more accurate than those computed using uninformative priors.

More broadly, the ability to synthesize background knowledge about a topic of interest into well-calibrated Bayesian priors is a prerequisite for deploying LMs in high-stakes environments ranging from portfolio construction to policy analysis. OPENESTIMATE reveals that current models fall well short of this bar, and we hope it serves as both a diagnostic tool and a catalyst for developing LMs that can function not just as knowledge retrievers, but as reliable probabilistic reasoners. To support future research and reproducibility, we release our code, benchmark dataset, and evaluation framework.[1]

## 2 THE OPENESTIMATE BENCHMARK

In this section, we describe the design of the OPENESTIMATE benchmark. We begin by describing our procedure for defining derived conditional statistics as estimation targets across domains (Section 2.1). We then explain how models are prompted to specify their priors by selecting and parameterizing probability distributions (Section 2.2). Finally, we outline the evaluation metrics used to assess the accuracy and calibration of the resulting priors (Section 2.3).

### 2.1 DERIVED CONDITIONAL STATISTICS AS ESTIMATION TARGETS

Evaluating LMs' probabilistic estimation skills requires asking questions for which the answers are known but don't appear in pretraining data. This is difficult to achieve in practice: most of human knowledge is contained in pretraining corpora, so coming up with new questions would require collecting new data experimentally, which is costly and time-consuming. OPENESTIMATE sidesteps this problem by defining *derived conditional statistics*: summary statistics computed by filtering large-scale observational datasets on specific conditions and aggregating over a target attribute. Because the filtering conditions are chosen randomly, the resulting statistics are empirically verifiable yet unlikely to correspond to well-documented facts in pretraining corpora.

We begin by selecting existing data sources spanning three broad domains: social sciences (Glassdoor[2], labor economics), industrial settings (Pitchbook(PitchBook Data, 2024), finance), and medicine (NHANES(Centers for Disease Control and Prevention, 2018), public health).

---

[1]https://github.com/alanarenda/openestimate
[2]https://www.kaggle.com/datasets/thedevastator/jobs-dataset-from-glassdoor

| Domain | Dataset | # marginal | # 1 cond | # 2 cond | # 3 cond | Total | Example |
|---|---|---|---|---|---|---|---|
| Labor Economics | Glassdoor | 1 | 16 | 20 | 6 | 43 | Midpoint salary |
| Finance | Pitchbook | 4 | 17 | 20 | 20 | 61 | Total funding |
| Human Health | NHANES | 14 | 20 | 20 | 20 | 74 | Total cholesterol |

Table 1: Distribution of statistics across domains. Columns indicate the number of marginal statistics and conditional statistics with one, two, or three conditioning attributes.

From each dataset, we construct two types of statistics. *Marginal statistics* are computed using every row of the dataset—- in NHANES, for example, this looks something like "the mean weight of adults in the US". *Conditional statistics*, on the other hand, are computed over filtered subsets of this dataset where the filtering is done by applying up to three additional conditions—- for instance, "the mean weight of adults (1) who a diabetes diagnosis, (2) who take medication for depression, and (3) have cholesterol above a given threshold.

We sample these conditions at random from the empirically observed values in the dataset. Following Xia et al. (2024), we require that each additional ccondition to shift the marginal statistic by at least 5%, ensuring that the statistics reflect meaningful variation across subpopulations rather than minor fluctuations due to sampling noise.

The full generation procedure is described in Algorithm 1 and depicted in Figure 1. A summary of the generated statistics for each domain are reported in Table 1.

---

**Algorithm 1:** Sampling $N_k$ *marginal* ($k = 0$) and *conditional* ($k = 1, 2, 3$) statistics

**Input:** data $D$, auxiliary attributes $\mathcal{A}$, counts $\{N_k\}_{k=0}^3$, threshold $\tau$, $n$ minimum sample size
**Output:** set $\mathcal{V}$ of statistics
$\mathcal{V} \leftarrow \emptyset, \mathcal{S} \leftarrow \emptyset$            `// S tracks which attributes have already been used`
**for** $k \in \{0, 1, 2, 3\}$ **do**
    **while** number of variables in $\mathcal{V}$ with $k$ attributes $< N_k$ **do**
        sample $k$ *distinct* attributes $\mathbf{a}_k \subset \mathcal{A}$          `// aₖ is a set of k attributes`
        $D' \leftarrow$ filter $D$ by $\mathbf{a}_k$          `// keep rows matching attributes in aₖ`
        **if** $|D'| < n$ **then**
            **continue**          `// skip if filtered sample is too small`
        $\mu^* \leftarrow$ mean$[d_v : d \in D']$          `// estimate mean on D'`
        $se^* \leftarrow$ SE$(\mu^*; D')$          `// estimate standard error on D'`
        $\mu_0 \leftarrow$ mean$[d_v : d \in D]$          `// unconditional mean on full D`
        **if** $|\mu^* - \mu_0| > \tau$ **and** $|\mu^* - \mu_0| > se^*$ **and** $\mathbf{a}_k \notin \mathcal{S}$ **then**
            add $(\mathbf{a}_k, \mu^*, se^*)$ to $\mathcal{V}$          `// store valid statistics`
            add $\mathbf{a}_k$ to $\mathcal{S}$          `// store attributes to avoid reuse`
**return** $\mathcal{V}$

---

While some of the resulting statistics may be contained within pretraining corpora (for example, the overall diabetes prevalence in the United States is a widely reported on figure), many others are far less likely to have been explicitly documented. In particular, the conditional statistics— for example, "the mean weight of adults with diabetes who are over 40, have elevated cholesterol, and take medication for depression," or "the median deal size for companies in a specific sector with a given number of employees" — represent random combinations of attributes that are unlikely to have been reported on directly. That said, a model with strong domain knowledge should still be able to reason toward accurate estimates—for instance, by drawing on adjacent knowledge of how diabetes prevalence varies with age, or how deal sizes differ across industries. By randomly sampling these conditions, we generate a large set of estimation targets that remain grounded in real-world observational data yet avoid testing shallow factual recall.

## 2.2 SPECIFYING BELIEFS AS BAYESIAN PRIORS

How should we elicit LM estimates about the likely values of these statistics? The simplest approach would be to prompt models to produce point estimates, but these capture only first-order accuracy—whether the estimate is close to the true value—and say nothing about whether the model has appropriate uncertainty about its answer. By instead requiring estimates to be expressed as Bayesian

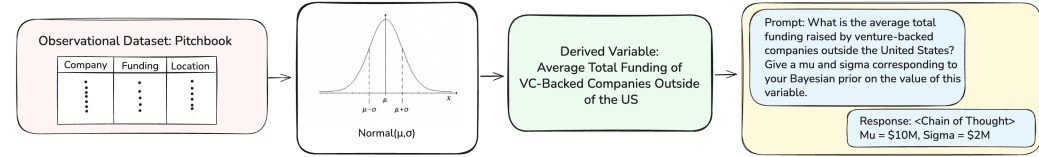

Figure 1: Statistic generation and prior elicitation pipeline. We construct derived statistics from large-scale observational datasets (e.g., PitchBook), specify them as estimation targets, and prompt language models to provide Bayesian priors on their true values.

priors (probability distributions over the variable of interest), we can evaluate both the accuracy of the model's central estimate and the calibration of its uncertainty. Thus, in OPENESTIMATE, we instead require estimates to be specified as Bayesian priors (probability distributions on the variable of interest).

Models are provided with a brief natural language description of the statistic of interest and are instructed to select and parameterize the functional form of the target distribution accurately. For all experiments in this paper, the models selected either a Gaussian, Beta, or log-normal distribution to parameterize:

$$X \sim \mathcal{N}(\mu, \sigma^2), \quad X \sim \text{Beta}(\alpha, \beta), \quad \text{or} \quad X \sim \text{LogNormal}(\mu, \sigma^2),$$

We hypothesize that these three forms are chosen by LMs because they arise frequently in our domains of interest—Gaussians for continuous, symmetric quantities like wages; Betas for proportions like disease rates; and log-normals for right-skewed quantities like startup valuations. These distributions represent the models' Bayesian prior or belief about the true value of the statistic and can be combined with samples from the underlying dataset to produce posterior distributions.

## 2.3 EVALUATION METRICS

Given a prediction from the LM in the form of a Bayesian prior, how should we evaluate its quality? We focus on two complementary dimensions of performance:

- **Accuracy**: The degree to which the model assigns high probability density to regions close to the empirical ground-truth value.

- **Calibration**: The consistency between the model's stated uncertainty and empirical frequencies. A model is well-calibrated if events assigned probability $p$ occur with long-run frequency $p$, such that nominal coverage levels of prediction intervals match their realized coverage.

### 2.3.1 ACCURACY

To assess accuracy, we evaluate whether the model places the mean of its predicted prior close to the ground-truth value of the statistic.

To quantify this, we first compute the **mean absolute error (MAE)** between the mean of the prior, $\hat{p}_i(\mu)$, and the empirical ground-truth value $\mu_i^*$ estimated from the full dataset for each of the $n$ variables in the dataset:

$$\text{MAE}_{\text{LLM}} = \frac{1}{n} \sum_{i=1}^{n} |\mu_i^* - \text{mean}(\hat{p}_i)| \ .$$

To interpret these errors across statistics with different units, we report LM predictions relative to a statistical baseline derived from a small number of samples from the true distribution pulled from the underlying dataset. Starting from naïve flat priors ($\alpha = 1, \beta = 1$ for Beta distributions; $\mu = 0, \sigma^2 = 10^5$ for Gaussians), we draw a random sample $\tilde{D}$ of size $|\tilde{D}| = 5$ from the relevant sub-population ($D'$ in Algorithm 1, corresponding to a sample of e.g. 5 patients or 5 job postings), from which we can compute a posterior $\tilde{p}_i(\mu \mid \tilde{D})$.

We then compute the statistical baseline MAE as the expected error across such samples:

$$\text{MAE}_{\text{baseline}} = \mathbb{E}_{\tilde{D}} |\mu_i^* - \text{mean}(\tilde{p}_i(\cdot \mid \tilde{D}))| \ .$$

We summarize performance using the error ratio, defined as the LM's MAE relative to this baseline:

$$\text{Error Ratio} = \frac{\text{MAE}_{\text{LLM}}}{\text{MAE}_{\text{baseline}}}.$$

An error ratio below one indicates that the LM's prior is more accurate than a small, noisy sample from the population whose properties are being estimated.

We also consider the **win rate** of the LLM prior to the statistical baseline, which is the percentage of the time that the model's estimate is closer to the ground truth than the statistical baseline.

$$\text{Win Rate (LLM prior > baseline)} = \frac{1}{N} \sum_{i=1}^{N} \mathbf{1}\{\text{MAE}_{\text{LLM},\,i} < \text{MAE}_{\text{baseline},\,i}\}.$$

In addition to the $N = 5$ baseline used for computing MAEs, we report win rates against baselines with varying numbers of samples.

Finally, we evaluate the usefulness of these priors in combination with data by computing an **LLM posterior**:

$$\hat{\tilde{p}}(\mu \mid \tilde{D}) \propto \hat{p}(\mu)\, p(\tilde{D} \mid \mu) \tag{1}$$

(as in the statistical baselines, but replacing the naïve prior with $\hat{p}$). As with priors, we evaluate the win rate of these posteriors relative to statistical baselines.

Together, these two dimensions provide a more complete picture of accuracy: the error ratio tests the average error of models relative to the statistical baselines whereas the win rate determines how consistently the LLMs outperform the baselines.

### 2.3.2 CALIBRATION

A model is well-calibrated if the probabilities it assigns correspond to empirical frequencies: events predicted to occur with probability $p$ should occur about $p$ percent of the time. In our setting, this means that the ground-truth value should fall into each predicted quantile with the correct long-run frequency.

To measure this, we compute the **continuous ranked probability score (CRPS)**, which penalizes both miscalibrated predictions and overly dispersed distributions. CRPS measures the distance between the predicted cumulative distribution function $F$ and the ground truth $y$ without binning:

$$\text{CRPS}(F, y) = \int_{-\infty}^{\infty} \left( F(x) - \mathbb{I}(x \geq y) \right)^2 dx$$

where $\mathbb{I}(x \geq y)$ is the indicator function. Lower values indicate better predictive performance.

As with MAE, we compare LM performance to a statistical baseline computed from small samples, where $\tilde{p}_i(\cdot \mid \tilde{D})$ is the posterior distribution obtained from a sample $\tilde{D}$ of size $|\tilde{D}|$:

$$\text{CRPS}_{\text{baseline}} = \mathbb{E}_{\tilde{D}} \left[ \frac{1}{n} \sum_{i=1}^{n} \text{CRPS}(\tilde{p}_i(\cdot \mid \tilde{D}), \mu_i^*) \right]$$

We then report the CRPS ratio:

$$\text{CRPS Ratio} = \frac{\text{CRPS}_{\text{LLM}}}{\text{CRPS}_{\text{baseline}}}$$

## 3 EVALUATION

Our evaluation is divided into two parts. In Section 3.1, we evaluate the zero-shot performance of current language models under standard inference settings. In Section 3.2, we zero in on the best-performing models and analyze how changing inference-time settings like system prompt, temperature, and prior elicitation method affect prediction quality.

Figure 2: MAE error ratio of LLM prior to a naive statistical baseline computed using a uninformative prior and five examples from the true distribution. Most models are no better than five examples; some are significantly worse. There isn't a statistically significant gap in performance between most model families.

## 3.1 ZERO-SHOT EVALUATION

In this section, we focus on zero-shot performance under standard inference settings. We do not apply fine-tuning, retrieval augmentation, or prompt engineering beyond directly asking the model to parameterize the distribution of a variable. To contextualize the LMs' performance, we compare to four statistical baselines that use $N \in [5, 10, 20, 30]$ examples that are computed using the procedure described in Section 2.3.1.

We evaluate six state-of-the-art language models, including three reasoning models [3]: Meta Llama 3.1 8B, Meta Llama 3.1 70B (Grattafiori et al., 2024), OpenAI GPT-4 (Achiam et al., 2023), OpenAI o3-mini (OpenAI, 2025a), OpenAI o4-mini (OpenAI, 2025b), and Qwen3-235B-A22B (Yang et al., 2025). We exclude Llama 3.1 8B after it fails to correctly interpret units. We evaluate each model at a medium temperature or reasoning effort—corresponding to 0.5 for GPT-4, "medium" for o3-mini and o4-mini, 0.5 for Llama 3.1 70B Instruct Turbo, and 0.6 for Qwen3-235B-A22B. We use a standard system prompt and prior elicitation prompt which are described in full in Appendix A.1.

| Domain | Sample Size | % Prior Better | % Posterior Better |
|---|---|---|---|
| **Glassdoor** | 5 | 37.0% | 71.4% |
| | 10 | 21.7% | 69.0% |
| | 20 | 13.0% | 68.1% |
| | 30 | 8.7% | 70.5% |
| **Pitchbook** | 5 | 50.8% | 69.6% |
| | 10 | 50.8% | 76.5% |
| | 20 | 49.2% | 80.1% |
| | 30 | 50.8% | 81.6% |
| **NHANES** | 5 | 74.3% | 70.4% |
| | 10 | 59.5% | 65.1% |
| | 20 | 47.3% | 56.6% |
| | 30 | 37.8% | 50.4% |

Table 2: Win rate of the LLM prior relative to an $N$-sample statistical baseline, and win rate of an LLM posterior (LLM prior + $N$ samples) relative to a statistical baseline (uninformative prior + $N$ samples).

**Accuracy.** Fixing the model to o4-mini, we compare the win rate of LLM priors against statistical baselines computed using $N \in \{5, 10, 20, 30\}$ data points sampled from the true distribution. This addresses the question: "how many data samples is the LLM prior worth?" We then form LLM posteriors by updating the LLM prior with the same $N$ examples used to compute each baseline, and compare

---

[3] Here, reasoning models are defined as models that have undergone a dedicated training step that involves reinforcement learning for chain-of-thought.

the win rate of the LLM posterior against the corresponding statistical baselines. This comparison asks whether starting from an LLM prior leads to better posteriors than starting from an uninformative prior.

As shown in Table 2, we find that the standalone LLM priors generally outperform the five-sample baseline in ~40-70% of cases, though win rates rapidly drop off with larger numbers of samples. However, even though these priors are often inaccurate in isolation, they can be effectively combined with data: LLM posteriors outperform or match statistical baselines with naive priors.

Next, we compare the accuracy of different model families across domains by evaluating MAE relative to the five-sample statistical baseline. The results are shown in Figure 2. We find relatively little variation between most models (with the exception of Llama-70B), and that again, most models have average errors that are no better than five examples; some are significantly worse. This suggests that while the LM priors are often consistently better than the statistical baseline, they are worse in terms of average absolute error. On the whole, these results suggest that OPENESTIMATE is challenging for frontier models.

**Calibration.** Next, we assess model calibration.[4] Larger models and reasoning models tend to be better calibrated than smaller, non-reasoning models, but again, no single model family consistently outperforms the rest; specific rankings are domain dependent.

| Model | Glassdoor | NHANES | Pitchbook |
|---|---|---|---|
| GPT-4o | 3.31 | 1.86 | 1.10 |
| Llama-3-70B | 4.56 | 2.76 | 1.13 |
| Llama-3-8B | 10.56 | 19.17 | 2.74 |
| Qwen3-235B | 2.50 | 1.65 | 1.04 |
| o3-mini | 3.17 | 1.35 | 0.99 |
| o4-mini | 2.42 | 1.17 | 1.01 |

Table 3: CRPS Ratio by Model Family Across Domains (vs. 5-Sample Baseline)

Table 3 presents CRPS ratios comparing each model family to the five-sample baseline. Reasoning models (o3-mini and o4-mini) achieve the best overall performance. Performance varies considerably by domain: in Pitchbook, all models perform comparably to the baseline, while in NHANES, smaller models struggle significantly: Llama-3-8B performs 20 times worse than the baseline. Overall, model size and reasoning capabilities appear most critical in the NHANES domain, while even smaller models achieve reasonable performance in Pitchbook.

---

[4]We exclude the statistical baselines from Figure 3 in this analysis because the baselines derive their posteriors from the same dataset used to compute the ground-truth values. Therefore, larger sample sizes produce extremely tight distributions centered on the ground-truth mean, which leads the ground truth to almost always fall in the middle quartiles (e.g., second or third).

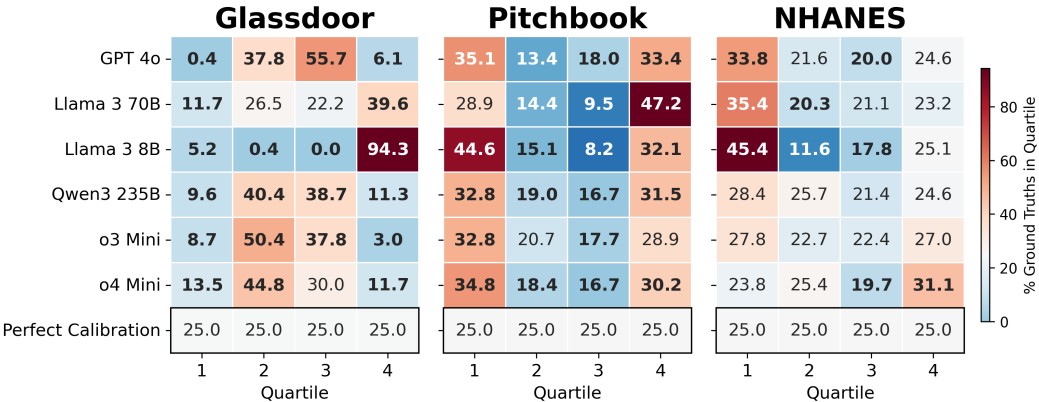

Figure 3: Heatmap describing the deviations from perfect calibration of each approach. Bolded values are statistically significant according to a per-quartile binomial test ($p < 0.05$). All approaches systematically overestimated across domains (Quartile 1 is greater than 25%). In some instances, there was high rates of both over and under-estimation (Quartile 1 and 4 are greater than 25%).

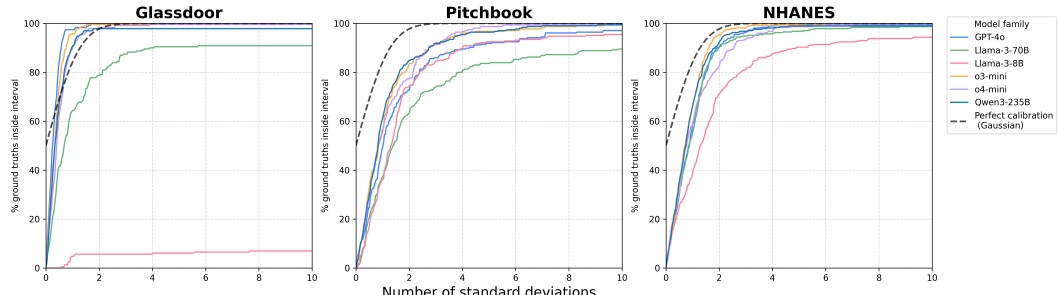

Figure 4: Cumulative distribution function displaying the percentage of ground truth values that fall within $n\sigma$ standard deviations away from the mean of the prior, where $\sigma$ is the standard deviation of the prior. The dashed line represents perfect calibration for a Gaussian. The best performing models have 80% of the ground truths within 1-2.5 standard deviations from the prior mean. There is overconfidence in Pitchbook and NHANES but underconfidence in Glassdoor.

We study whether this miscalibration is systematic across model families. As shown in Figure 3, we find that all model families exhibit a tendency towards systematic overestimation. In Pitchbook, overestimation is compounded by high rates of underestimation as well, with both tails overweighted.

Next, we evaluate how tightly models concentrate their uncertainty by examining the cumulative distribution of ground-truth values relative to the predicted priors (Figure 4). We find the best models cover 80% of the ground truth values within two to three standard deviations of the mean. However, performance is domain-dependent: in Glassdoor and NHANES, the best models cover over 80% of ground-truth values within two standard deviations, while in Pitchbook, three standard deviations are required. This suggests that even the strongest models vary substantially in how they express uncertainty across domains.

Finally, we analyze whether model-reported uncertainty is a reliable guide to predictive accuracy (Figure 5) by comparing the standard deviation ratio to the error ratio. Ideally, models are low error and well-calibrated. In the Glassdoor domain, models appear reasonably well-calibrated relative to the five-sample statistical baseline, but are consistently less accurate than this baseline. In contrast, models in Pitchbook are consistently more confident and less accurate than this baseline. Results in NHANES fall in between these extremes: models generally achieve lower error than in Glassdoor, but their uncertainty estimates are less well-calibrated, with several models exhibiting either under- or over-dispersion. Taken together, these results indicate that the relationship between uncertainty and accuracy is once again strongly domain-dependent.

We also assess whether predictive uncertainty aligns with accuracy by examining the rank correlation between the two for each model family. A stronger correlation between predictive uncertainty and accuracy would indicate that uncertainty is a good indicator of accuracy. However, the reality is mixed: uncertainty is a good indicator of accuracy in NHANES but not necessarily in Pitchbook or Glassdoor.

### 3.2 ABLATIONS

We investigate how inference-time settings influence the quality of elicited priors, focusing on three factors: (i) temperature or reasoning effort, (ii) system prompt, and (iii) elicitation protocol. To isolate their effects, we evaluate both a reasoning model (OpenAI o4-mini) and a non-reasoning model (OpenAI gpt-4o). The full set of results is shown in Appendix A.2. None of the settings tested has a consequential impact on performance, indicating that more sophisticated approaches to improving accuracy and calibration are needed.

## 4 RELATED WORK

Our work intersects with three major lines of language model research: evaluating probabilistic reasoning as a mathematical skill, structuring probabilistic reasoning for better estimation, and applications to forecasting.

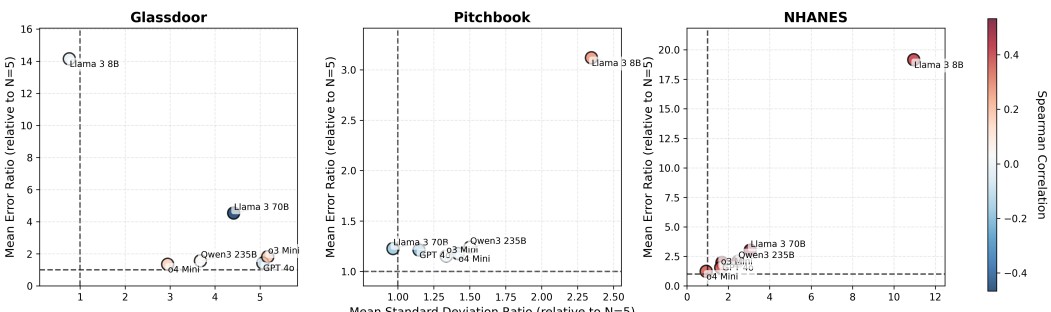

Figure 5: Relationship between uncertainty and accuracy across domains. Each point shows a model's error ratio versus its standard deviation ratio relative to the $N = 5$ baseline. Colors indicate the Spearman correlation between predictive uncertainty and accuracy within a single model's predictions, addressing the question of whether a given model tends to be comparatively more confident when it's more accurate. These correlations differ more so by domain than by model.

**Evaluating probabilistic reasoning.** One line of research examines how well LMs perform at problem-solving tasks involving structured probabilistic models. For example, Paruchuri et al. (2024) evaluate models' probabilistic reasoning given simple idealized distributions; Nafar et al. (2025) tests models' ability to provide probabilistic estimates given a Bayesian network; and Jin et al. (2023) examine the models' causal reasoning given probabilities. Collectively, these studies frame probabilistic reasoning as a mathematical exercise with clearly defined inputs and well-specified outputs. By contrast, our benchmark targets real-world estimation problems, where the relevant information must be inferred rather than provided and the ground truth itself may be ambiguous or unavailable.

**Structuring probabilistic reasoning.** Another line of work proposes structures for LM-based probabilistic reasoning to improve performance. Using "guesstimation" questions similar to ours, Xia et al. (2024) prompt LMs to propose relevant random variables and moment constraints, and then fits a log-linear distribution that satisfies these constraints. Feng et al. (2024) take a similar approach, and evaluate a multi-step process in which LMs brainstorm relevant factors, make coarse probabilistic assessments, and construct an approximate Bayesian network for inference. Huynh et al. (2025) use LLMs to generate synthetic counterfactual outcomes by sampling pseudo-observations, constructing empirical distributions. These approaches extend beyond single-variable reasoning by introducing latent structure and explicit intermediate steps. However, the focus for Xia et al. (2024) and Feng et al. (2024) is answering discrete multiple-choice questions, while Huynh et al. (2025) focuses on augmenting small datasets for downstream causal inference tasks rather than directly evaluating the quality of LLM-generated distributions.

Like our approach, Selby et al. (2025) elicit parametric Bayesian priors from LLMs. However, they evaluate priors by comparing them to human expert elicitation in existing psychology studies or to historical observational data in specific settings (e.g., precipitation and temperature in particular cities in December). By contrast, we specifically construct derived variables—complex aggregations and cross-sections of tabular data—across diverse domains; we directly evaluate accuracy and calibration relative to estimated ground truth; and we systematically evaluate how model family and inference-time settings impact results.

**Language model-based forecasting.** Recent studies have also evaluated LMs' forecasting capabilities (Karger et al., 2024; Halawi et al., 2024; Ye et al., 2024; Chang et al., 2024; Schoenegger et al., 2025). These works also test whether models can synthesize heterogeneous evidence into well-calibrated estimates, but they focus on making predictions about real-world future events. In contrast to our benchmark, the outcomes of forecasting questions are, by design, highly likely to appear in LMs' training data after they resolve; they thus perpetually become "stale" and must be replaced with new questions, as noted by Karger et al. (2024). By focusing on questions that require reasoning about fine-grained cross of tabular datasets, rather than future events, OPENESTIMATE questions are designed to remain challenging over time.

## 5 LIMITATIONS AND FUTURE WORK

While OPENESTIMATE provides a first step toward evaluating uncertainty in open-domain estimation, several limitations remain that point to directions for future work. Ground truth values in OPENESTIMATE were estimated from finite samples, and therefore might exhibit estimation error. Moreover, while OPENESTIMATE was constructed to reduce systematic information leakage, leakage still can occur to varying degrees. In terms of scope, the current benchmark is limited to questions derived from three datasets across three domains; expanding to new domains would lead to a more thorough evaluation of priors. In terms of evaluation, we focus our attention on zero-shot methods without retrieval or fine-tuning; studying training-time interventions for uncertainty awareness and domain adaptation would be a complementary next step in future work.

## 6 CONCLUSION

We introduced OPENESTIMATE, a benchmark and evaluation framework for assessing language models on open-ended probabilistic estimation with real-world tabular data. The benchmark (i) defines a realistic task where models must express beliefs as full probability distributions (Bayesian priors), (ii) elicits priors from the LMs through several protocols, and (iii) evaluates performance in terms of accuracy and calibration against statistical baselines. By focusing on derived conditional statistics from domains such as public health, labor economics, and finance, OPENESTIMATE probes the ability for models to reason under uncertainty while limiting direct factual lookup and information leakage.

## ACKNOWLEDGMENTS

Research was sponsored by the Department of the Air Force Artificial Intelligence Accelerator and was accomplished under Cooperative Agreement Number FA8750-19-2-1000. The views and conclusions contained in this document are those of the authors and should not be interpreted as representing the official policies, either expressed or implied, of the Department of the Air Force or the U.S. Government. The U.S. Government is authorized to reproduce and distribute reprints for Government purposes notwithstanding any copyright notation herein.

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

# A   APPENDIX A

## A.1   ZERO-SHOT ESTIMATION

We tested Llama 3 8B but excluded it from our analysis because it incorrectly followed instructions pertaining to units and had an average error that was orders of magnitude larger than the other models due to this mistake.

**System Prompt.**

```
Glassdoor
You are a helpful assistant that can answer questions about the labor market.
```

```
Pitchbook
You are a helpful assistant.
```

```
NHANES
You are a helpful assistant that can answer questions about human health.
```

## A.2   ABLATIONS

**Elicitation Protocol.**

```
Direct
You are a statistical
    expert tasked with constructing a prior distribution for a variable. Your goal
    is to choose the most appropriate distribution type and estimate its parameters.

Your estimates should reflect uncertainty about
    the population-level parameter, not the variation across individual observations.

Here is the variable you need to model:

{{variable}}

{{units_description}}

Available Distribution Types: Normal (Gaussian), Lognormal, Beta

Instructions:

1. Reasoning: First, provide detailed reasoning
    explaining how you arrived at your specific parameter values. Address: What range
    do you expect the population parameter to fall in and why? How certain/uncertain
    are you about this parameter? How do your chosen parameter values translate
    to meaningful quantities in the original scale (e.g., median, mean, quantiles,
    credible intervals)? Why is this distribution type appropriate for this variable?

2. Output: After your reasoning, provide
    your answer using EXACTLY these XML tags based on which distribution you choose:

If you choose Normal:
<distribution_type>Normal</distribution_type>
<mu>value</mu>
<sigma>value</sigma>

If you choose Lognormal:
<distribution_type>Lognormal</distribution_type>
```

```
<mu>value</mu>
<sigma>value</sigma>

CRITICAL: mu and sigma are parameters in LOG-SPACE, not real-space!

Key relationships to real-space values:
- MEDIAN (real-space) = exp(mu)
- MEAN (real-space) = exp(mu + sigma^2/2)
- MODE (real-space) = exp(mu - sigma^2)

How to set mu: First decide what you think
     the MEDIAN value should be (in the original units), then set mu = ln(median).
     Examples: If median should be 30 dollars, then mu = ln(30) = 3.4 approximately.
     If median should be 100 employees, then mu = ln(100) = 4.6 approximately.
     If median should be 1000 dollars, then mu = ln(1000) = 6.9 approximately.

How to set sigma:
     sigma controls the spread in log-space (typical values: 0.2 to 1.0). sigma = 0.3
     gives roughly a 95 percent credible interval of [exp(mu-0.6), exp(mu+0.6)]. sigma
     = 0.5 gives roughly a 95 percent credible interval of [exp(mu-1.0), exp(mu+1.0)].

Common mistake to avoid:

WRONG: Setting mu = 30 when you
     mean the value is 30 dollars (This gives median = exp(30) = 10 trillion dollars!)

CORRECT: Setting mu = ln(30) = 3.4 approximately
     when you mean the value is 30 dollars (This gives median = exp(3.4) = 30 dollars)

Always verify: Calculate exp(mu). Does this match
     your expected median? Calculate exp(mu + sigma^2/2). Does this match your expected
     mean? If these are wildly different from what you expect, you have made an error!

If you choose Beta:
<distribution_type>Beta</distribution_type>
<alpha>value</alpha>
<beta>value</beta>

Critical Unit Check: Pay close attention to units. If the variable says in millions
     USD, you need to work in millions! For example, I think the typical company
     has raised about 3.5 million dollars. In millions, this is: 3.5, NOT 3500000!

Now, please analyze the variable
     and provide your reasoning followed by your distribution choice and parameters.
```

## Quantile

```
You are a statistical expert tasked with constructing a prior distribution
     for a variable. Your goal is to choose the most appropriate distribution type and
     express your uncertainty about the parameters true value using quantile estimates.

Your estimates should reflect uncertainty about
     the population-level parameter, not the variation across individual observations.

Here is the variable you need to model:

{{variable}}

{{units_description}}

Available Distribution Types:
```

```
Normal (Gaussian):
     For variables that can be positive or negative, symmetric around the mean

Lognormal:
     For strictly positive variables, often right-skewed (e.g., prices, sizes, counts)

Beta: For variables bounded between 0 and 1 (e.g., proportions, probabilities)

Instructions:

1. Consider the context of the variable,
     including its meaning and any relevant information that informs your beliefs.

2. Choose the
     most appropriate distribution type based on: The natural bounds of the variable
     (can it be negative? is it bounded between 0 and 1?). The expected shape of
     uncertainty (symmetric vs. skewed?). The nature of the quantity being estimated.

3. Estimate the following percentiles of the parameters true
     value: 5th percentile (only a 5 percent chance the true value is below this). 25th
     percentile. 50th percentile (median, your best estimate of the true value). 75th
     percentile. 95th percentile (only a 5 percent chance the true value is above this).

4. Begin your analysis by showing your thought process inside
     <parameter_estimation_process> tags. Include the following elements: Explicitly
     state the type of parameter being estimated (e.g., population mean, proportion).
     Explain why you chose a particular distribution type. List any known facts or data
     points about the variable. Consider and list possible data sources or methods for
     estimating this parameter. Brainstorm factors that might influence the parameters
     value. Note potential biases or limitations in the available information.
     State any assumptions you are making. Consider how the parameter might have
     changed over time or across different subgroups. Provide your quantile estimates
     with a brief explanation for each. Include relevant facts or context about
     the variable. Justify your choices. Emphasize population parameter uncertainty
     (not individual variability). Reflect on what your estimate spread indicates
     about your certainty. Consider any plausible edge cases or alternative scenarios.

5. After your analysis, provide your final answer in the following format:

<distribution_type>[Normal, Lognormal, or Beta]</distribution_type>
<q5>[5th percentile value]</q5>
<q25>[25th percentile value]</q25>
<q50>[50th percentile (median) value]</q50>
<q75>[75th percentile value]</q75>
<q95>[95th percentile value]</q95>

<justification>
[Brief summary of your reasoning, including why you chose this distribution type]
</justification>

<confidence_level>
[Description of how certain or uncertain you are, and why]
</confidence_level>

Examples:

1. Normal Distribution Example:
Variable: Average temperature in a city during summer
Units: Degrees Celsius

<distribution_type>Normal</distribution_type>
<q5>22</q5>
<q25>24</q25>
```

```
<q50>26</q50>
<q75>28</q75>
<q95>30</q95>

<justification>
Normal distribution is appropriate because temperature can
    be positive or negative and uncertainty about the mean is approximately symmetric.
    Based on historical climate data and considering year-to-year variation.
    The spread reflects uncertainty in long-term averages due to climate variability.
</justification>

<confidence_level>
Moderately confident. Climate data is well-documented,
    but climate change introduces some uncertainty about current averages.
</confidence_level>

2. Lognormal Distribution Example:
Variable: Average home price in a metropolitan area
Units: Thousands of USD

<distribution_type>Lognormal</distribution_type>
<q5>280</q5>
<q25>350</q25>
<q50>420</q50>
<q75>520</q75>
<q95>680</q95>

<justification>
Lognormal distribution is appropriate
    because home prices are strictly positive and typically right-skewed. Based on
    recent market data and regional economic indicators. The asymmetric spread (wider
    on the high end) reflects the possibility of higher prices in desirable areas.
</justification>

<confidence_level>
Somewhat uncertain. Housing markets are volatile and
    influenced by many factors including interest rates and local economic conditions.
</confidence_level>

3. Beta Distribution Example:
Variable: Proportion of customers who complete a purchase after adding items to cart
Units: Proportion (0 to 1)

<distribution_type>Beta</distribution_type>
<q5>0.55</q5>
<q25>0.62</q25>
<q50>0.68</q50>
<q75>0.74</q75>
<q95>0.80</q95>

<justification>
Beta distribution is appropriate
    because this is a proportion bounded between 0 and 1. Based on industry benchmarks
    for e-commerce conversion rates and typical cart abandonment patterns. The spread
    accounts for variation across different product categories and customer segments.
</justification>

<confidence_level>
Moderately confident. Conversion rates are well-studied
    in e-commerce, but can vary significantly by industry and website design.
</confidence_level>
```

```
Critical Unit Check: Pay close attention to units. If the variable says in millions
    USD, you need to work in millions. For example, I think the typical company
    has raised about 3.5 million dollars. In millions, this is 3.5, not 3500000.

Remember to tailor
    your analysis to the specific variable and units provided, focusing on uncertainty
    about the population-level parameter rather than individual variability.
```

## Mean-Variance

```
You are a statistical expert tasked with constructing
    a prior distribution for a variable. Your goal is to choose the most appropriate
    distribution type and estimate its parameters using mean and standard deviation.

Your estimates should reflect uncertainty about
    the population-level parameter, not the variation across individual observations.

Here is the variable you need to model:

{{variable}}

{{units_description}}

Available Distribution Types:

Normal (Gaussian):
    For variables that can be positive or negative, symmetric around the mean

Lognormal:
    For strictly positive variables, often right-skewed (e.g., prices, sizes, counts)

Beta: For variables bounded between 0 and 1 (e.g., proportions, probabilities)

Instructions:

1. Consider the context of the variable, including what it
    represents and any relevant information or assumptions that inform your beliefs.

2. Choose the
    most appropriate distribution type based on: The natural bounds of the variable
    (can it be negative? is it bounded between 0 and 1?). The expected shape of
    uncertainty (symmetric vs. skewed?). The nature of the quantity being estimated.

3. Estimate the following quantities:
    Best guess (mean): your estimate of the most likely value of the population-level
    parameter. Standard deviation: a numerical expression of your uncertainty
    about the true value, not the variability across individual observations.

4. Begin your analysis by showing your thought process
    inside <parameter_estimation_process> tags. Include the following elements:

Clearly state
    the type of parameter being estimated (e.g., population mean, true proportion).
    Explain why you chose a particular distribution type. List any known facts, data
    points, or previous estimates about the variable. Consider possible data sources,
    analogous populations, or related studies that inform your belief. Identify
    key factors that might influence the value of the parameter. Note any limitations,
    uncertainties, or assumptions in your reasoning. Reflect on how the parameter
    might differ across subgroups or change over time. Provide your best guess
    (mean) and your estimate of the standard deviation. Justify your choices with
    reference to the context, data, and assumptions. Emphasize that your uncertainty
    pertains to the population parameter, not individual variation. Reflect on what
```

```
        the magnitude of your standard deviation implies about your confidence. Consider
        plausible edge cases or outliers that helped you calibrate your uncertainty.

5. After your analysis, provide your final answer in the following format:

<distribution_type>[Normal, Lognormal, or Beta]</distribution_type>
<mean>[Best guess for the true value]</mean>
<std_dev>[Standard deviation representing your uncertainty]</std_dev>

<justification>
[Brief summary of your reasoning, including
     why you chose this distribution type and what informed your parameter estimates]
</justification>

<confidence_level>
[Explanation of how confident or uncertain you are, and why]
</confidence_level>

Examples:

1. Normal Distribution Example:
Variable: Average height of adult males in a country
Units: Centimeters

<distribution_type>Normal</distribution_type>
<mean>175</mean>
<std_dev>2.5</std_dev>

<justification>
Normal distribution is appropriate because
     height can theoretically take any value and is approximately symmetric around
     the mean. Based on global averages, previous studies in similar populations,
     and considering factors like nutrition and genetics. The standard deviation
     reflects uncertainty due to potential sampling biases and regional variations.
</justification>

<confidence_level>
Moderately confident. While height is
     well-studied, variations between regions and over time introduce some uncertainty.
</confidence_level>

2. Lognormal Distribution Example:
Variable: Average annual revenue of small businesses in a region
Units: Thousands of USD

<distribution_type>Lognormal</distribution_type>
<mean>250</mean>
<std_dev>150</std_dev>

<justification>
Lognormal distribution
     is appropriate because revenue is strictly positive and typically right-skewed,
     with some businesses earning significantly more than the median. Based on industry
     reports and regional economic data. The standard deviation reflects substantial
     uncertainty due to variation across industries and economic conditions.
</justification>

<confidence_level>
Somewhat uncertain. Business revenue varies widely by industry
     and economic conditions, and available data may not be fully representative.
</confidence_level>

3. Beta Distribution Example:
```

```
Variable: Proportion of people who prefer tea over coffee in a city
Units: Proportion (0 to 1)

<distribution_type>Beta</distribution_type>
<mean>0.6</mean>
<std_dev>0.05</std_dev>

<justification>
Beta distribution is appropriate because this is a proportion bounded
    between 0 and 1. Estimated based on local cultural preferences, limited survey
    data, and comparison with similar cities. The standard deviation accounts for
    potential biases in available data and variations across different demographics.
</justification>

<confidence_level>
Somewhat uncertain. Beverage preferences can vary significantly based on
    factors like age, cultural background, and local trends, which are not fully known.
</confidence_level>

Critical Unit Check: Pay close attention to units. If the variable says in millions
    USD, you need to work in millions. For example, I think the typical company
    has raised about 3.5 million dollars. In millions, this is 3.5, not 3500000.

Remember: you are
    modeling beliefs about the parameter, not the spread of raw data. Your standard
    deviation should reflect how much uncertainty you have about the single true
    value that governs the population, not the spread of outcomes across individuals.

Provide your analysis and final answer based on the given variable and
    units description. Your final output should consist only of the formatted answer
    and should not duplicate or rehash any of the work you did in the thinking block.
```

**Additional Results.**

We calculate expected calibration error as a complimentary metric for calibration. Let $Q_{ij}$ be the $j$-th quartile bin of $\hat{p}_i$. We define $\hat{q}_j = \frac{1}{n} \sum_{i=1}^{n} \mathbf{1}\{\mu_i^* \in Q_{ij}\}$. Formally, we compute the **quartile expected calibration error (ECE)** as:

$$\text{ECE} = \sum_{j=1}^{4} |\hat{q}_j - 0.25| .$$

Lower values indicate better calibration, with ECE $= 0$ corresponding to perfect calibration (at quartile granularity).

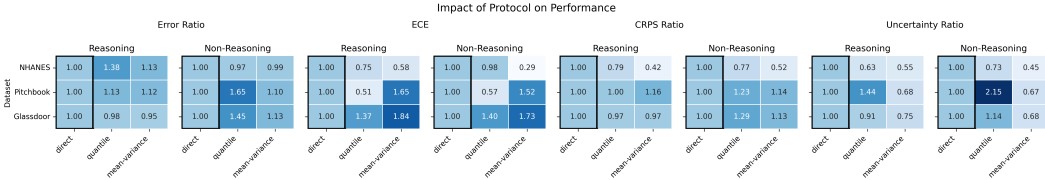

Figure 6: Effect of elicitation protocol (direct, quantile, mean–variance) on error ratio, expected calibration error (ECE), CRPS ratio, and uncertainty (standard deviation) across reasoning and non-reasoning models, relative to direct elicitation.

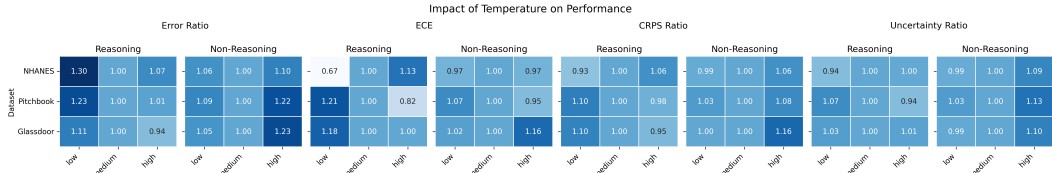

Figure 7: We examine the impact of changing temperature or reasoning effort on accuracy, calibration, and certainty.

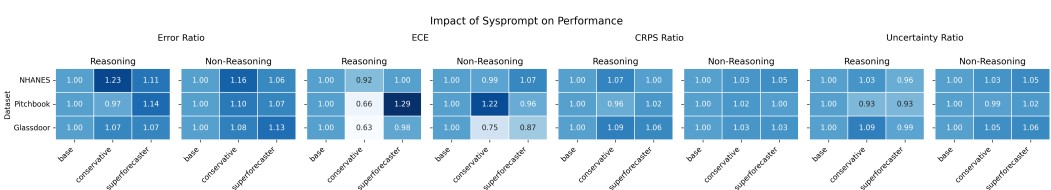

Figure 8: We examine the impact of changing the system prompt or reasoning effort on accuracy, calibration, and certainty.

