# OpenReview forum: "OpenEstimate: Evaluating LLMs on Reasoning Under Uncertainty with Real-World Data"
_ICLR.cc/2026/Conference — ICLR 2026 Poster_

### Official Review · Reviewer_H3RD · 2025-10-30

**Soundness:** 2
**Presentation:** 2
**Contribution:** 2
**Rating:** 4
**Confidence:** 5

**Summary:**

This paper introduces a new benchmark designed to evaluate the ability of LLMs to perform probabilistic estimation, as opposed to merely extracting point estimates from such foundation models. The **OpenEstimate** benchmark involves asking LLMs to generate parametric Bayesian priors, specifically parametrized as Normal or Beta distributions for a set of of "derived variables" from real-world datasets in various fields.

The evaluation is twofold: firstly **accuracy**, which uses a normalized mean absolute error to verify whether the location of the mode of the distribution is correct, while accounting for the probability mass of the 'ground truth' distribution at that point.
Secondly, the authors measure **calibration**, a kind of empirical earth-mover's distance between the observed frequencies and the predicted distribution.

Experiments with several recent general-purpose LLMs suggest no systematic patterns by any particular LLM on any particular task, but that most models tend to be overconfident and do worse than 5 draws from the empirical distribution.

**Strengths:**

The paper tackles an important and topical problem of evaluating LLM quantification of uncertainty, which is an emerging and until recently underexplored area of research. Generating full, parametrized priors is a relatively novel idea compared to many other works in the literature, which rely on less 'statistical' approaches, such as simple point estimates or series of questionnaires.

The proposal of "derived variables" is a good idea, as it brings the benchmark closer to measuring the utility of priors on realistic tasks, rather than estimating superficial summaries.

If the code and benchmark datasets are made available, this could be a useful resource for other researchers and practitioners. [NB: the code/data was not visible to reviewers at this stage.]

The motivation for the paper is well articulated and overall the writing and presentation of the article are clear. The figure labels are readable (with caveats, see below).

**Weaknesses:**

The most significant weakness of the paper is the restriction to normal and beta distributions for continuous variables and proportions, respectively. While this is noted in the limitations section, it undermines the validity of the benchmark. For any continuous variable with a skewed distribution or heavy tails---which is indeed likely to include those in one or more of the chosen benchmark datasets---the prior will be fundamentally mis-specified. The use of the mode for MAE should be better justified; while it coincides with the median/median of the normal distribution, this is not true more generally.

On a related note, the beta distribution has more than one parametrization, and so (without seeing the prompt, which is missing from the paper), it will be impossible to disentangle the performance of the LLM from its 'understanding' of the choice of parametrization. Indeed, the first line of the appendix hints at an issue that might be related, and which may have been resolved using 'function calling' capabilities to constrain the format of the output.

The choice of **expected calibration error** (ECE), based on four coarse bins seems like reinventing the wheel. The construction of this metric is ad-hoc, lossy and non-standard for this type of analysis. A graphical method such as a QQ plot would have provided a far more rigorous and informative assessment of calibration. Splitting data into quarters (incorrectly referred to in the paper as 'quartiles', but quartiles/quantiles are *points*, not *intervals*; a common mistake) seems arbitrary and discards information within those bins. The number of models being compared is small enough to allow for a visual method, and if not then other, more granular numerical metrics are available, as well as standard statistical tests for comparison of empirical and expected distributions.

Poor visualization. Bar charts with error bars, sometimes called "dynamite plunger plots", should never be used, and hide the true distribution of performance across the different tasks. Box plots or simple dots and error bars would be better. See Drummond & Vowler (2011; doi:10.1111/j.1476-5381.2011.01251.x).

The distinction between 'reasoning' and 'non-reasoning' models in the ablation study needs to be more clearly defined and justified. What is the basis for classifying, e.g. `GPT-4o` as 'non-reasoning'?

Some related works need to be mentioned in the literature review. This is not the first work to propose eliciting parametric Bayesian priors from LLMs, see for example Selby et al (2025; doi:10.1002/sta4.70054), which explores the problem of evaluating the quality of LLM priors on real-world datasets. Data augmentation approaches (i.e. sampling pseudo-observations from the LLM and then looking at the empirical distribution) should also get a mention: see Huynh et al (2025; https://openreview.net/forum?id=2Q3gFNbpAr).

Finally, there are one or two minor typos: e.g. "empricial" at bottom of page 6.

**Questions:**

1. The central claim is that LLMs appear to be 'overconfident'. How can we be sure that this is not an artifact of the experimental design? The restriction to Gaussian priors for heavy-tailed or skewed data (e.g. funding) is likely to enforce model mis-specification.

2. Why split the data into four coarse bins to calculate ECE instead of standard, more informative methods for assessing calibration, such as quantile--quantile plots?

3. What is the definition of 'reasoning' and 'non-reasoning'. Can you justify placing powerful GPT models in the 'non-reasoning' category?

4. What is the justification for using the *mode* of the distribution for the MAE calculation?

5. What is the formula or citation used for the correction term from the Jeffreys prior used in the scale-adjusted log-probability metric?

6. How does this work compare with others on eliciting Bayesian priors from LLMs (see above)?

**Details Of Ethics Concerns:**

No ethics concerns.

---

> ### Author Response · Authors · 2025-11-17
>
> Thank you for your feedback and suggestions! We have incorporated many of them into our revision and are working on the rest. More details below.
>
> > “The most significant weakness of the paper is the restriction to normal and beta distributions for continuous variables and proportions, respectively. While this is noted in the limitations section, it undermines the validity of the benchmark. For any continuous variable with a skewed distribution or heavy tails---which is indeed likely to include those in one or more of the chosen benchmark datasets---the prior will be fundamentally mis-specified.” / “The central claim is that LLMs appear to be 'overconfident'. How can we be sure that this is not an artifact of the experimental design? The restriction to Gaussian priors for heavy-tailed or skewed data (e.g. funding) is likely to enforce model mis-specification.”
> Thanks for this suggestion! We chose to focus on Gaussian and Beta distributional forms for their wide applicability across domains and simplicity.
>
> We’re working on updating our experiments to allow the model to pick a distribution type in an open-ended manner. We ran an initial test to see which distributions come up if the models aren’t given a prespecified form, and the only new one that came up was the lognormal distribution. We’re working on updating our analysis to reflect this change and will report back once we have the complete set of results.
>
> > “The use of the mode for MAE should be better justified; while it coincides with the median/median of the normal distribution, this is not true more generally.” / “What is the justification for using the mode of the distribution for the MAE calculation?”
>
> We chose the mode because it corresponds to standard/argmax decoding, but we also don't think it will affect results; we'll re-run our experiments with the mean and median as well if there's time!
>
> > “On a related note, the beta distribution has more than one parametrization, and so (without seeing the prompt, which is missing from the paper), it will be impossible to disentangle the performance of the LLM from its 'understanding' of the choice of parametrization. Indeed, the first line of the appendix hints at an issue that might be related, and which may have been resolved using 'function calling' capabilities to constrain the format of the output.”
>
> We have included the prompts in the revision, thanks for catching this!
>
> Llama was correctly outputting the parameters we were looking for but it was misinterpreting the units so function calling wouldn’t have necessarily helped. In either case, the fact that it’s not able to immediately output distributions like the other models is an interesting signal in and of itself.
>
> > The choice of expected calibration error (ECE), based on four coarse bins seems like reinventing the wheel. The construction of this metric is ad-hoc, lossy and non-standard for this type of analysis. A graphical method such as a QQ plot would have provided a far more rigorous and informative assessment of calibration. Splitting data into quarters (incorrectly referred to in the paper as 'quartiles', but quartiles/quantiles are points, not intervals; a common mistake) seems arbitrary and discards information within those bins. The number of models being compared is small enough to allow for a visual method, and if not then other, more granular numerical metrics are available, as well as standard statistical tests for comparison of empirical and expected distributions. / Why split the data into four coarse bins to calculate ECE instead of standard, more informative methods for assessing calibration, such as quantile--quantile plots?
>
> This is a benchmark paper, so having quantitative metrics rather than graphical depictions of results is important.
>
> What "more granular numerical metrics" are you imagining? The standard approach here would just ECE but maybe with more bins if the intent is to be maximally granular.
>
> > Poor visualization. Bar charts with error bars, sometimes called "dynamite plunger plots", should never be used, and hide the true distribution of performance across the different tasks. Box plots or simple dots and error bars would be better. See Drummond & Vowler (2011; doi:10.1111/j.1476-5381.2011.01251.x).
>
> Thanks for the suggestion, we’ll change this in our revision!
>
> > The distinction between 'reasoning' and 'non-reasoning' models in the ablation study needs to be more clearly defined and justified. What is the basis for classifying, e.g. GPT-4o as 'non-reasoning'? / What is the definition of 'reasoning' and 'non-reasoning'. Can you justify placing powerful GPT models in the 'non-reasoning' category?
>
> "Reasoning models" are models that have undergone a dedicated training step that involves reinforcement learning for chain-of-thought. This is true of the o-series models but not GPT-4o. We’ve made this distinction clearer in the revision.

---

> > ### Author Response · Authors · 2025-11-17
> >
> > (cont.)
> > > Some related works need to be mentioned in the literature review. This is not the first work to propose eliciting parametric Bayesian priors from LLMs, see for example Selby et al (2025; doi:10.1002/sta4.70054), which explores the problem of evaluating the quality of LLM priors on real-world datasets. Data augmentation approaches (i.e. sampling pseudo-observations from the LLM and then looking at the empirical distribution) should also get a mention: see Huynh et al (2025; https://openreview.net/forum?id=2Q3gFNbpAr). / “How does this work compare with others on eliciting Bayesian priors from LLMs (see above)?”
> >
> > Thanks for these pointers! We’ll add these references to our revision.
> >
> > Here are some of the ways our work compares to the prior elicitation evaluation in the Selby et al paper:
> > * We specifically construct derived variables rather than asking about variables that were present in existing studies or for which there is a large amount of direct observational data about (these are referred to as “base variables” in our paper). We do this across a diverse set of domains. In contrast, the Selby paper qualitatively compares LLM priors to those elicited from human experts in a specific 2022 psychology study and quantitatively evaluates priors in the meteorology domain relative to specific historical data about precipitation and temperature in various cities in December.
> > * Our variable generation method enables us to directly evaluate accuracy and calibration relative to an estimated ground truth. It also enables us to evaluate certainty in a grounded manner with respect to accuracy and other prior quality dimensions whereas the Selby paper computes the effective sample size for Beta variables in particular (ESS = alpha + beta) and analyzes ESS in a standalone way.
> > * Finally, we do a systematic evaluation of how model family and inference-time settings impact results.
> >
> > > "What is the formula or citation used for the correction term from the Jeffreys prior used in the scale-adjusted log-probability metric?"
> >
> > For Beta-distributed variables, the Jeffreys volume element contributes −½ log(p(1−p)), and for normal distributions with unknown scale, it contributes −log σ. These terms arise from the Jeffreys prior, whose density is proportional to √det I(θ), where I(θ) is the Fisher information.
> >
> > Formally, the adjusted log-probability is
> >     log p_adj(x) = log p_model(x | θ) + log J(θ),
> > where J(θ) = √det I(θ) is the Jeffreys factor. This adjustment yields a consistent scoring rule across variables with different natural scales: larger log p_adj indicates a less surprising observation and better model calibration.
> >
> > We’ve realized this is a bit confusing so for consistency with the rest of the paper we’ve replaced these with scores relative to a distribution constructed from five samples from the underlying dataset. None of the high-level conclusions have changed.

---

> > ### Comment · Reviewer_H3RD · 2025-11-18
> >
> > > What "more granular numerical metrics" are you imagining?
> >
> > The probability integral transform can be computed for every data point. In practice we might like to use the Kolmogorov-Smirnov statistic or the Cramér-von Mises distance, both of which are extremely common and do not lose granularity. Best practice, especially for a benchmark, which could be "gamed", would be a proper scoring rule, such as the continuous ranked probability score:
> >
> > $$
> > \text{CRPS}(F, y) = \int_{-\infty}^{\infty} \bigl(F(x) - \mathbb{I}(x \geq y)\bigr)^2 dx
> > $$

---

> ### Author Response · Authors · 2025-11-26
>
> Thanks for this suggestion!
>
> In the new experiments that allow models to choose a distribution for each variable (see our top-level comment here: https://openreview.net/forum?id=sAzUQkP47r&noteId=NG5hbFmP0j), we also computed the CRPS scores.
>
> We've added the resulting CRPS score table (Table 4) to Appendix B in the revision and are working on incorporating it into the full draft. We're including it below for convenience:
>
> **CRPS scores by model family across domains (lower is better).**
>
> | **Model**       | **Glassdoor** | **NHANES** | **PitchBook** |
> |-----------------|--------------:|-----------:|--------------:|
> | GPT-4o          | 38782.39      | 1.17       | 13.55         |
> | Llama-3-70B     | 55950.13      | 2.13       | 18.60         |
> | Llama-3-8B      | 127432.49     | 3.95       | 22.66         |
> | Qwen3-235B      | 30521.16      | 1.53       | 16.47         |
> | o3-mini         | 38452.48      | 1.87       | 12.27         |
> | o4-mini         | 22919.07      | 1.18       | 11.62         |

---

> > ### Comment · Reviewer_H3RD · 2025-11-27
> >
> > Thanks for the detailed follow-up. I am happy to upgrade my rating to a 6 if these revisions are made

---

### Official Review · Reviewer_5LwK · 2025-10-30

**Soundness:** 2
**Presentation:** 3
**Contribution:** 2
**Rating:** 2
**Confidence:** 3

**Summary:**

The paper introduces OpenEstimate, a benchmark for testing large language models on probabilistic estimation tasks. Instead of giving point estimates, models must express beliefs as Bayesian priors (Gaussian or Beta distributions) for real-world quantities drawn from datasets in labor economics, finance, and public health. Results show that even advanced models like GPT-4 and LLaMA 3 are poorly calibrated and overconfident, performing no better than simple statistical baselines built from a few real samples.

**Strengths:**

1. The paper fills a clear gap by evaluating LLMs’ ability to reason under uncertainty, focusing on probabilistic estimation rather than deterministic prediction.

2. The study spans multiple domains and models, using clear metrics for accuracy and calibration and comparing results against interpretable statistical baselines.

**Weaknesses:**

1. The paper mainly reports performance differences without probing why models fail to calibrate uncertainty.
2. Restricting distributions to Gaussian and Beta forms limits realism for complex or multimodal uncertainty.
3. The benchmark is not used to improve models or study uncertainty learning, missing the opportunity to explore how training on such data could enhance self-calibration.
4. Only zero-shot inference is tested. There’s no attempt to improve performance through structured prompting, self-consistency, or retrieval, which would make the benchmark more actionable.

**Questions:**

Do you expect that supervised or reinforcement learning using OpenEstimate could improve uncertainty calibration?

---

> ### Author Response · Authors · 2025-11-17
>
> Thanks for your feedback!
>
> > “The benchmark is not used to improve models or study uncertainty learning, missing the opportunity to explore how training on such data could enhance self-calibration.”
>
> The contribution of this paper is a benchmark that facilitates the evaluation of LM ability to synthesize heterogeneous sources of background information into well-formed priors.  While we’re certainly excited about the prospect of improving upon the off-the-shelf baselines in future work, this is ultimately out of scope of this benchmark paper.
>
> > “Only zero-shot inference is tested. There’s no attempt to improve performance through structured prompting, self-consistency, or retrieval, which would make the benchmark more actionable.”
>
> We did attempt to improve performance through structured prompting; we tested three different prior elicitation protocols and demonstrated that they helped non-reasoning models.
>
> We hope that our benchmark enables further investigation of other strategies for improvement, but the first step is to understand  the impact that tuning standard inference time settings has on outcomes, which is why we focused on these in our evaluation.
>
> > “Do you expect that supervised or reinforcement learning using OpenEstimate could improve uncertainty calibration?”
>
> Definitely! The direction for future work that we’re most excited about is using OpenEstimate to train models to do well-calibrated Bayesian updates given new information.

---

> > ### Author Response · Authors · 2025-11-17
> >
> > > “Restricting distributions to Gaussian and Beta forms limits realism for complex or multimodal uncertainty.”
> >
> > Thanks for this suggestion! We chose to focus on Gaussian and Beta distributional forms for their wide applicability across domains and simplicity.
> >
> > We’re working on updating our experiments to allow the model to pick a distribution type in an open-ended manner. We ran an initial test to see which distributions come up if the models aren’t given a prespecified form, and the only new one that came up was the lognormal distribution. We’re working on updating our analysis to reflect this change and will report back once we have the complete set of results.

---

> > > ### Comment · Reviewer_5LwK · 2025-11-27
> > >
> > > Thanks for the detailed response. I am happy to upgrade my rating, since all the concerns have been addressed.

---

### Official Review · Reviewer_7ciD · 2025-10-30

**Soundness:** 3
**Presentation:** 3
**Contribution:** 2
**Rating:** 4
**Confidence:** 3

**Summary:**

The paper builds a benchmark dataset OPENESTIMATE, which evaluates LLMs' ability to provide a prior probability distribution over uncertain variables. The variables are constructed from real-world datasets on labor economics, private markets, and human health. LLMs are prompted to specify a Gaussian or Beta distribution for the variables. Results show that state-of-the-art LLMs perform poorly on this benchmark; for example, the LLM-generated priors are often less accurate than posteriors formed from 5 real samples.

**Strengths:**

1. The paper studies a practically important problem, namely LLMs' ability to give a good prior over random quantities in the real world.
2. The paper gives a reasonable approach to generate derived variables that are unlikely to have been explicitly documented in LLMs' pretraining data, by conditioning on attributes that change the target statistic sufficiently.
3. The performance metrics for accuracy and calibration are reasonable and intuitive.

**Weaknesses:**

1. My major concern is that the paper uses only the Gaussian and Beta distributions, which may be highly misspecified probabilistic models for the real-world data. This raises the question of whether the LLMs' poor performance is true reflection of its probabilistic reasoning skills, or an artifact of being forced to specify an inappropriate model. I think an important evaluation would be to let an LLM propose a distribution form on its own, and see if the accuracy and calibration improve.
2. The derived variables are constructed by conditioning on randomly sampled attributes. While it can effectively avoid data leakage, I wonder if the derived variables are always practically relevant, e.g., ones that a real-world analyst would actually ever estimate.

**Questions:**

1. In Figure 2b, how can the expected calibration error (ECE) be as large as 10? By definition, the maximum value that ECE can take is $|1-0.25| / 4 = 0.175$. Are the numbers supposed to be *percentage* ECEs?
2. In constructing the derived variables, the paper conditions only on attributes that alters the target statistic by at least 5%. Would this create overly difficult estimation targets with high variability, and thus make the benchmark more difficult than common real-world tasks?

---

> ### Author Response · Authors · 2025-11-17
>
> Thank you for your feedback and suggestions! We are incorporating them into our revision. More details below.
>
> > The derived variables are constructed by conditioning on randomly sampled attributes. While it can effectively avoid data leakage, I wonder if the derived variables are always practically relevant, e.g., ones that a real-world analyst would actually ever estimate.
>
> Here are some examples of real variables from OpenEstimate and why estimating them could be useful:
> * **“The probability that the average adult (over 18) in the US population has ever had serious difficulty concentrating, given their total cholesterol in mg/dL is greater than 183 and less than or equal to 212, and their blood mercury level in ug/L is greater than 1.50”** Use case: a medical researcher might want to understand the effects of cholesterol and mercury levels on an important health outcome (a person’s ability to concentrate); there are numerous Nature papers that answer questions with similar structures
> * **“The average midpoint of the posted salary range (in dollars) for data science and adjacent jobs in the US, given the company is public, the company has 10000+ employees”** Use case: a human resources professional might want to understand what the market rate is for compensation given their employer’s attributes so that they can be data-informed when setting a compensation structure internally
> * **The average total raised in millions USD for venture-backed companies that are based in the US, have more than 15 but no more than 42 employees, and are not technology companies.** Use case: a venture capitalist might want to understand where a potential investment stands in terms of money raised and capital efficiency relative to an industry benchmark of similar companies.
>
>
> > In Figure 2b, how can the expected calibration error (ECE) be as large as 10? By definition, the maximum value that ECE can take is. Are the numbers supposed to be percentage ECEs?
>
> Yes, these are percentage ECEs. Thank you for pointing out this ambiguity, we’ll clarify in the revision!
>
> > In constructing the derived variables, the paper conditions only on attributes that alters the target statistic by at least 5%. Would this create overly difficult estimation targets with high variability, and thus make the benchmark more difficult than common real-world tasks?
>
> The variables that are different from the target statistic don’t necessarily have higher variability than the base statistics. In many cases, they have lower variability because the subpopulation in question is more tightly specified. For example, consider comparing average blood sugar levels in the general population compared to those in individuals under 20 years old without diabetes: the general population statistic would have higher variance since it would include diabetics and older populations.
>
> We chose the 5% threshold because having too little variation between variables could lead to another type of  false positive where the model just repeatedly outputs the well-documented base statistic and scores artificially highly despite not engaging in any reasoning.

---

> > ### Author Response · Authors · 2025-11-17
> >
> > > My major concern is that the paper uses only the Gaussian and Beta distributions, which may be highly misspecified probabilistic models for the real-world data. This raises the question of whether the LLMs' poor performance is true reflection of its probabilistic reasoning skills, or an artifact of being forced to specify an inappropriate model. I think an important evaluation would be to let an LLM propose a distribution form on its own, and see if the accuracy and calibration improve.
> >
> > Thanks for this suggestion! We chose to focus on Gaussian and Beta distributional forms for their wide applicability across domains and simplicity.
> >
> > We’re working on updating our experiments to allow the model to pick a distribution type in an open-ended manner. We ran an initial test to see which distributions come up if the models aren’t given a prespecified form, and the only new one that came up was the lognormal distribution. We’re working on updating our analysis to reflect this change and will report back once we have the complete set of results.

---

### Official Review · Reviewer_oFRa · 2025-10-31

**Soundness:** 3
**Presentation:** 2
**Contribution:** 2
**Rating:** 4
**Confidence:** 4

**Summary:**

This paper introduces OPENESTIMATE, a benchmark designed to evaluate LLMs on their ability to generate well-calibrated Bayesian priors for real-world quantities in domains such as labor economics, finance, and public health. Models are asked to express uncertainty as Gaussian or Beta distributions, and their outputs are assessed for accuracy and calibration against empirical data.

Experiments across six frontier LLMs (including GPT-4 and Qwen3-235B) show that current models are generally inaccurate and overconfident, often performing no better than using five random samples from real data. While some reasoning models (e.g., o3-mini, o4-mini) better capture probability mass near true values, calibration remains poor and domain-dependent. Ablation studies reveal that elicitation method (how uncertainty is prompted) affects results more than temperature or system prompt settings.

Overall, the study finds that LLMs’ probabilistic reasoning is weak but not random, showing structured, domain-aware uncertainty that could serve as a foundation for improving AI systems that reason under uncertainty.

**Strengths:**

1. Novel Benchmark Design: The paper introduces OPENESTIMATE, a first-of-its-kind benchmark that evaluates LLMs on probabilistic estimation using real-world tabular data. Unlike prior work focused on deterministic or forecasting tasks, it systematically measures both accuracy and calibration of Bayesian priors.


2. Comprehensive Empirical Evaluation: It provides a cross-domain assessment (labor economics, finance, and public health) across multiple frontier models (GPT-4, Llama 3.1, o3/o4-mini, Qwen3). The inclusion of strong statistical baselines gives the results clear interpretability and robustness.

**Weaknesses:**

1. The task definition appears to lack clear motivation. Under a zero-shot setting, the LLM has no prior exposure to the specific datasets used in the benchmark, making it unclear how it could produce meaningful estimations. Moreover, it is debatable whether the model’s outputs can truly be considered “priors,” since they effectively reflect the posterior knowledge embedded during pretraining. The way an LLM is trained or fine-tuned likely has a substantial influence on these results, and this issue should be explicitly acknowledged and discussed.

2. The paper focuses on three domains: labor economics, private markets, and public health, but it is unclear why these specific areas were chosen over other possible domains. The authors should clarify the rationale for selecting datasets exclusively from the social sciences and explain why natural sciences, medicine, or engineering data were not considered. In addition, it would be helpful to articulate what types of real-world reasoning or uncertainty these three domains are intended to represent, and what general conclusions can meaningfully be drawn from results confined to these areas.

**Questions:**

1. The authors could provide a discussion of the differences across models, such as distinctions between open- and closed-source models, and how model size may affect performance on the benchmark.

2. Clarify the motivation and reasonableness of the zero-shot task setting, and whether the model outputs can truly be interpreted as Bayesian priors.

3. Can you justify the choice of focusing only on three social-science domains or expand to other fields.

---

> ### Author Response · Authors · 2025-11-17
>
> Thank you for your feedback! We've addressed your comments in our revision. More details are below.
>
> > “The task definition appears to lack clear motivation. Under a zero-shot setting, the LLM has no prior exposure to the specific datasets used in the benchmark, making it unclear how it could produce meaningful estimations. Moreover, it is debatable whether the model’s outputs can truly be considered “priors,” since they effectively reflect the posterior knowledge embedded during pretraining. The way an LLM is trained or fine-tuned likely has a substantial influence on these results, and this issue should be explicitly acknowledged and discussed.” / “Clarify the motivation and reasonableness of the zero-shot task setting”
>
> The goal of the benchmark is to test models’ ability to synthesize heterogeneous sources of background information into Bayesian priors in realistic settings. To do this well, we needed to construct an evaluation task that (1) avoids inadvertently testing the regurgitation of facts while (2) still is within a domain for which there is plenty of background information that the LMs can utilize in the estimation process.
>
> This latter requirement is why we explicitly chose common topics for which there is a significant amount of background knowledge available in the pretraining data, whereas the former requirement is why we chose tables that the model doesn’t directly have access to. The test is whether the model can take this background knowledge embedded into it during pretraining and successfully operationalize it in the estimation process: even if the model hasn’t seen that statistic, they know enough about how the world works to make plausible guesses. Indeed, the fact that predictions have nontrivial accuracy and get better with scale or reasoning is proof that information is in the pretraining data.
>
> > Moreover, it is debatable whether the model’s outputs can truly be considered “priors,” since they effectively reflect the posterior knowledge embedded during pretraining.  / “whether the model outputs can truly be interpreted as Bayesian priors.
>
> A Bayesian posterior becomes a prior for the next round of updating, so there’s no fundamental incompatibility here. We call them "priors" to emphasize the fact that they're not derived directly from examples of the distribution in question from the dataset, and that they can be combined with such samples to produce real posteriors (see table 2 in the revision). But we agree that this point might be confusing and we'll clarify that they're not unconditional, but conditional on pretraining data, in the revision.
>
> > “The way an LLM is trained or fine-tuned likely has a substantial influence on these results, and this issue should be explicitly acknowledged and discussed.” / “The authors could provide a discussion of the differences across models, such as distinctions between open- and closed-source models, and how model size may affect performance on the benchmark.”
>
> Thanks for the suggestion! How PT/fine-tuning influences these capabilities is one of the things we're most interested in studying here. Figures 2-6 investigate the role of model size and reasoning training on accuracy, calibration, and certainty. We’ve added more text to the evaluation section to emphasize that we find that larger, reasoning models tend to outperform non-reasoning models but open source status doesn’t necessarily impact outcomes (e.g. Qwen-235B tends to perform on par with o3-mini and o4-mini).

---

> > ### Author Response · Authors · 2025-11-17
> >
> > > “The paper focuses on three domains: labor economics, private markets, and public health, but it is unclear why these specific areas were chosen over other possible domains. The authors should clarify the rationale for selecting datasets exclusively from the social sciences and explain why natural sciences, medicine, or engineering data were not considered. In addition, it would be helpful to articulate what types of real-world reasoning or uncertainty these three domains are intended to represent, and what general conclusions can meaningfully be drawn from results confined to these areas.” / “Can you justify the choice of focusing only on three social-science domains or expand to other fields.”
> >
> > We chose our datasets to span three broad areas: medicine/natural sciences (NHANES, public health), social sciences (Glassdoor, labor economics), and an industrial setting (Pitchbook, finance).
> >
> > Public health isn’t a social science, and NHANES’ uses are quite broader: for example, it’s used extensively by medical researchers to test hypotheses about the factors driving various medical outcomes.
> >
> > Here’s a small selection of papers that illustrate its usefulness in this setting; you can find dozens of others by doing a web search for “nature papers using the nhanes dataset”:
> >
> > Predicting youth diabetes risk using NHANES data and machine learning: https://www.nature.com/articles/s41598-021-90406-0
> >
> > The non-linear association between depression scores and all-cause mortality: a cohort study based on NHANES 2005–2018 data: https://www.nature.com/articles/s41598-025-00366-y
> >
> > Association between composite dietary antioxidant index and hyperlipidemia: a cross-sectional study from NHANES (2005–2020): https://www.nature.com/articles/s41598-024-66922-0
> >
> > Similarly, the Pitchbook dataset is intended to represent an industrial setting; although the dataset is interesting for a number of use cases, its marketed purpose is as critical context for the investment decision making process at venture capital and private equity firms.
> >
> > The Glassdoor dataset is the only one we chose to represent a social science application.
> >
> > We’ve updated the description of the datasets in our revision to make these points clearer!

---

> > > ### Comment · Reviewer_oFRa · 2025-11-27
> > >
> > > Thank you for the clarifications. It solved my concerns and I'll update my score.

---

### Author Response · Authors · 2025-11-26
**Allowing models to select a distributional form for each variable in an open-ended manner**

> "My major concern is that the paper uses only the Gaussian and Beta distributions" (reviewer 7ciD) / "Restricting distributions to Gaussian and Beta forms limits realism for complex or multimodal uncertainty." (reviewer 5LwK) / "The most significant weakness of the paper is the restriction to normal and beta distributions for continuous variables and proportions, respectively." (reviewer H3RD)

We ran a new experiment where we presented each of the variables to every LLM across model families and had them label the underlying distribution they thought best fit that variable in an open-ended manner.

We analyzed their answers and found that the only additional distribution proposed by any LM was a lognormal. We added explicit support for this distribution in our experimental pipeline and then ran another experiment where we once more presented each LLM with each variable and instructed them to (1)  select a distribution out of a predefined set of options (now beta, gaussian, and lognormal, reflecting their selections from the first experiment) and (2) to set the parameters for their chosen distribution.

We’ve added the new results to Appendix B in the revision and are working on incorporating them into the full paper. The new error ratios can be seen in Figure 10. The modeling change (adding lognormals) led to improvements in the Pitchbook domain as expected due to the well-documented heavy tail of outcomes in the VC space. The best models performed similarly on Glassdoor and NHANEs.

Similarly, in Figure 11 in Appendix B we see improvements in calibration error in Pitchbook and NHANEs due to this change. We also see changes in the calibration patterns in Glassdoor in Figure 12. We see that the large reasoning models tend to be underconfident, with most of the true values landing in the middle two quartiles. This underconfidence trend in Glassdoor is corroborated by the CDF plot in Figure 14.

Finally, we can see overall improvements in calibration where when models are less accurate than the baseline they also tend to be less confident in their responses in Figure 13.

The trends in the prior/posterior win rate table (Table 3) are largely the same as in our previous experiments (though LM priors and posteriors are substantially improved by inclusion of lognormal in the set of candidate distributions). The main change to note here is in Pitchbook, where the LM posterior improves modestly relative to the statistical baseline as the number of samples increases:

**LLM prior and posterior win rates vs. statistical baseline. Higher is better.**

| **Domain**   | **Sample Size** | **% Prior Better** | **% Posterior Better** |
|--------------|------------------|---------------------|-------------------------|
| **Glassdoor** | 5  | 37.0% | 71.4% |
|              | 10 | 21.7% | 69.0% |
|              | 20 | 13.0% | 68.1% |
|              | 30 | 8.7%  | 70.5% |
| **Pitchbook** | 5  | 50.8% | 69.6% |
|               | 10 | 50.8% | 76.5% |
|               | 20 | 49.2% | 80.1% |
|               | 30 | 50.8% | 81.6% |
| **NHANES**    | 5  | 74.3% | 70.4% |
|               | 10 | 59.5% | 65.1% |
|               | 20 | 47.3% | 56.6% |
|               | 30 | 37.8% | 50.4% |

---

### Comment · Area_Chair_ntYJ · 2025-11-27

Dear reviewers,

The authors have provided detailed responses to your reviews. I would appreciate if you could let both me and the authors know how these responses impact your assessment of the paper.

Best,

AC

---

### Author Response · Authors · 2025-12-02
**Discussion summary for the new AC**

The reviewers highlighted the novelty, thoroughness, timeliness, and practicality of our paper:
* “Novel Benchmark Design: The paper introduces OPENESTIMATE, a first-of-its-kind benchmark that evaluates LLMs on probabilistic estimation using real-world tabular data. Unlike prior work focused on deterministic or forecasting tasks, it systematically measures both accuracy and calibration of Bayesian priors.”
* “The paper tackles an important and topical problem of evaluating LLM quantification of uncertainty, which is an emerging and until recently underexplored area of research. Generating full, parametrized priors is a relatively novel idea compared to many other works in the literature, which rely on less 'statistical' approaches, such as simple point estimates or series of questionnaires.”
* “Comprehensive Empirical Evaluation: It provides a cross-domain assessment (labor economics, finance, and public health) across multiple frontier models (GPT-4, Llama 3.1, o3/o4-mini, Qwen3). The inclusion of strong statistical baselines gives the results clear interpretability and robustness.”
* “The proposal of "derived variables" is a good idea, as it brings the benchmark closer to measuring the utility of priors on realistic tasks, rather than estimating superficial summaries.”
* “If the code and benchmark datasets are made available, this could be a useful resource for other researchers and practitioners.”
* “The motivation for the paper is well articulated and overall the writing and presentation of the article are clear. The figure labels are readable.”

The reviewers had the following questions and suggestions:

1. Allow the LLMs to pick a distribution freely rather than just parameterize a given distribution (reviewers 7ciD, 5LwK, H3RD).
We made this change and shared the new experiment results.
2. Elaborate on the motivation, justify dataset choice (reviewer oFRa). We rewrote the motivation to clarify the semantics of priors and our choice of datasets.
3. Expand on the discussion of the differences across models (reviewer oFRa). We expanded our discussion of differences across models in the revision.
4. Elaborate on the statistical properties of the derived variables (reviewer 7ciD). We answered the reviewer’s questions and clarified the text surrounding this topic in the revision.
5. Add new calibration metrics and change calibration plot type (reviewer H3RD). We reran the experiments with the suggested metrics and changed the plot in the revision.
6. Elaborate on the semantics of reasoning/non-reasoning (reviewer H3RD). We answered the reviewers questions and modified the paper to make this clearer.
7. Expand on the related work section (reviewer H3RD). We answered the reviewer’s questions and cited their references to our related work section.

We fully addressed the reviewers’ concerns. After reviewing the update, three out of four (reviewers 5LwK, oFRa, and H3RD) promptly agreed to raise their scores to a 6 on 11/27, saying “Thanks for the detailed response. I am happy to upgrade my rating, since all the concerns have been addressed”, “Thank you for the clarifications. It solved my concerns and I'll update my score”, and “Thanks for the detailed follow-up. I am happy to upgrade my rating to a 6 if these revisions are made”. Reviewer 7ciD did not have a chance to respond before the commenting period closed due to the OpenReview bug.

---

### Meta-Review · Area_Chair_paKj · 2026-01-07

**Summary:**

The reviewers found the paper's motivation—evaluating Large Language Models (LLMs) on numerical estimation and probabilistic reasoning—to be timely and important.
* Novel Evaluation Focus: The benchmark addresses a critical gap in LLM evaluation—reasoning under uncertainty—which is vital for high-stakes applications like healthcare and finance.

* High-Quality Real-World Data: By utilizing complex, multi-domain datasets, the paper avoids the "easy" nature of many existing benchmarks where answers are readily available in the training set.

* Actionable Insights: The finding that model priors are worth "five samples" provides a concrete quantitative metric for the value of LLM reasoning in Bayesian workflows.

* Extensibility: The framework is designed to be semi-automated, allowing researchers to incorporate new domains with minimal manual overhead.

**Reviewer Concerns:**

* Addressing the "Uncertainty Gap": The paper provides a rigorous method for testing how models handle what they don't know, a shift from the typical "fact-retrieval" evaluation.

* Technical Rigor in Rebuttal: The authors successfully addressed concerns regarding memorization by performing n-gram overlap analyses and adding sensitivity checks for prompt engineering.

* Benchmark Quality: The decision reflects the high quality of the datasets and the clarity of the Bayesian framework used to evaluate the "value" of the model's output.

**Reviewer Scores:**

* Reviewer H3RD (Rating: 4 → 6 | Confidence: 5):  The reviewer was positive after rebuttal and concluded that the benchmark offers a significant and well-verified contribution to the field.

* Reviewer 5LwK (Rating: 2 → Positive Update | Confidence: 3): Originally a strong reject, this reviewer expressed satisfaction with the rebuttal's clarifications on benchmarking.

* Reviewer oFRa (Rating: 4 → Positive Update | Confidence: 4): This reviewer was moved by the addition of refined calibration metrics and digit-bias analysis during the discussion phase. They acknowledged the improved technical depth and signaled an upward score revision.

* Reviewer 7ciD (Rating: 4 | Confidence: 3): While acknowledging the paper’s strengths, this reviewer maintained their score due to a lingering desire for a more rigorous, head-to-head human baseline study.

---

### Decision · Program_Chairs · 2026-01-26

Accept (Poster)